# Phase transition kinetics of superionic H₂O ice phases revealed by Megahertz X-ray free-electron laser-heating experiments

R. J. Husband [1] ✉, H. P. Liermann [1], J. D. McHardy [2], R. S. McWilliams [2], A. F. Goncharov [3], V. B. Prakapenka [4], E. Edmund [3], S. Chariton[4], Z. Konôpková[5], C. Strohm[1], C. Sanchez-Valle [6], M. Frost [7], L. Andriambariarijaona[8], K. Appel[5], C. Baehtz [9], O. B. Ball [2], R. Briggs[10], J. Buchen [11,15], V. Cerantola [5,16], J. Choi [12], A. L. Coleman [10], H. Cynn [10], A. Dwivedi[5], H. Graafsma [1], H. Hwang [1,17], E. Koemets[11,18], T. Laurus [1], Y. Lee [12], X. Li[1,19], H. Marquardt [11], A. Mondal [6], M. Nakatsutsumi [5], S. Ninet[8], E. Pace [2], C. Pepin [13], C. Prescher[14], S. Stern[1,20], J. Sztuk-Dambietz[5], U. Zastrau [5] & M. I. McMahon[2] ✉

H₂O transforms to two forms of superionic (SI) ice at high pressures and temperatures, which contain highly mobile protons within a solid oxygen sublattice. Yet the stability field of both phases remains debated. Here, we present the results of an ultrafast X-ray heating study utilizing MHz pulse trains produced by the European X-ray Free Electron Laser to create high temperature states of H₂O, which were probed using X-ray diffraction during dynamic cooling. We confirm an isostructural transition during heating in the 26-69 GPa range, consistent with the formation of SI-bcc. In contrast to prior work, SI-fcc was observed exclusively above ~50 GPa, despite evidence of melting at lower pressures. The absence of SI-fcc in lower pressure runs is attributed to short heating timescales and the pressure-temperature path induced by the pump-probe heating scheme in which H₂O was heated above its melting temperature before the observation of quenched crystalline states, based on the earlier theoretical prediction that SI-bcc nucleates more readily from the fluid than SI-fcc. Our results may have implications for the stability of SI phases in ice-rich planets, for example during dynamic freezing, where the preferential crystallization of SI-bcc may result in distinct physical properties across mantle ice layers.

The high pressure ($P$) phase diagram of H₂O is remarkably complex, containing numerous phases that differ in their crystal structures[1–3], chemical bonding[4–6], and degree of orientational[7–9] and dynamical disorder[6,10]. In recent years, particular interest has been given to the high temperature ($T$), superionic (SI) forms of solid H₂O (ice)[11–15], which are characterized by the fast diffusion of protons within a crystalline oxygen sublattice. In particular, studies of the stability fields, structural behavior, and protonic conductivity of SI ice are motivated by the theorized presence of vast quantities of these phases in the interiors of Uranus and Neptune[16,17], where the unusual properties of these SI ice phases have been proposed to contribute to the non-dipolar and non-axisymmetric magnetic fields of these planets[18]. Since its initial

prediction[19], SI ice has been the focus of numerous theoretical studies which proposed that it adopts a body-centered cubic (SI-bcc) oxygen sublattice at low pressures[20–22], transforming to a face-centered (SI-fcc) lattice above ~100 GPa[23–25]. The SI-bcc oxygen lattice is isostructural with that of cubic ice VII, which is stable at lower temperatures, but the boundary between the two phases is first order in nature[26]. Experimental studies, however, are comparably scarce due to the challenges associated with creating and probing SI states using both static and dynamic compression methods. The stability field of SI ice cannot be reached by single shock compression of the ambient liquid due to the steep rise in temperature along the principal Hugoniot at low pressures, and the first experimental verification of SI ice was achieved using shock compression of pre-compressed samples[11] and a reverberating shock compression scheme[14,27,28] based on its high (total) conductivity (>100 S/cm)[27,28] and low optical conductivity («100 S/cm)[11] above 100 GPa. The crystal structure in the SI region (at 160 GPa and 3200 K) was first identified as fcc using nanosecond X-ray diffraction (XRD)[14], with the fcc phase remaining stable up to pressures >400 GPa; however, the subsequent observation of a bcc crystal structure at 3300 K and 5500 K at 205 GPa[29] suggests that the boundary between the bcc and fcc phases is strongly dependent on the time-dependent $P$-$T$ path, and may lie at higher pressures than previously anticipated.

Static compression experiments using diamond anvil cells (DACs) and synchrotron XRD have mapped out the $H_2O$ phase diagram at lower pressures. However, experiments conducted above ~1300 K, which are performed either by direct heating of $H_2O$ using a $CO_2$ laser (wavelength $\lambda$ ~ 10.6 μm) or by indirect heating of an embedded laser absorber which interacts with near IR radiation ($\lambda$ ~ 1 μm), can encounter additional complexities arising from temperature fluctuations (e.g. induced by changes in the optical properties of the sample during a phase transition[13]) or the presence of temperature gradients within the sample chamber[30]. Indirect laser heating experiments using laser couplers are particularly susceptible to temperature gradients, resulting in a discernible signal originating from the colder regions of the sample in XRD patterns. Although $CO_2$ laser heating provides more spatially uniform heating as the laser is directly absorbed by the sample, $CO_2$ lasers typically exhibit large power fluctuations at microsecond timescales due to the use of pulse width modulation to control the average laser power[31]. In addition, direct $CO_2$ laser heating of $H_2O$ is not possible above ~60 GPa due to the reduced absorption of $\lambda$ ~ 10 μm radiation above this pressure[12,32]. Consequently, despite extensive experimental efforts, the location of the melting curve and the SI-bcc/fcc boundary remain controversial, with recent studies reporting melting temperatures which exhibit discrepancies of ~1000 K at 50 GPa[13,15,32–34].

Ionic conductivity measurements performed using resistively-heated DACs initially identified the onset of superionic conduction at $T$ ~ 740 K at 56 GPa[35]; however, the maximum reported conductivity (0.12 S/cm) was ~2 orders of magnitude lower than values determined in shock compression experiments at higher pressures[27,28]. A subsequent XRD study reported the observation of an isostructural phase transition on compression above 15.6(2) GPa at 905 K[15], which was identified as the SI-bcc to ice VII transition based on the large entropy increase at the transition and agreement with results from ab initio molecular dynamics simulations. The same work also saw evidence of SI-bcc during laser heating in the 27–44 GPa pressure range, with the appearance of SI-bcc coinciding with the first observation of fluid diffraction[15]. In more comprehensive studies, the SI-bcc to SI-fcc transition was observed in two independent laser heating studies[12,13], but with significant discrepancies in the reported phases diagrams. In particular, Prakapenka et al.[13] observed SI-fcc during heating at pressures as low as ~30 GPa, whereas Weck et al.[12] did not observe evidence of SI-fcc below 50 GPa. Instead, Weck et al. observed fluid/SI-bcc coexistence on heating in the 27–47 GPa range at temperatures below the SI-bcc to SI-fcc transition temperatures reported by Prakapenka

et al. The SI-fcc phase boundaries reported by Prakapenka et al. consistently lie at higher temperatures than those reported by Weck et al., with reported SI-bcc/SI-fcc transition temperatures differing by ~600 K at 60 GPa.

In this work, the structural properties and phase stability region of SI ice were investigated using trains of MHz pulses produced by the European X-ray Free Electron Laser (European XFEL), which were utilized to create and probe high temperature states of statically-compressed $H_2O$. This experiment used a femtosecond serial pump-probe X-ray heating scheme, where each pulse deposits energy into the sample and simultaneously captures a snapshot of the sample as it cools from the hot state created by the previous pulse[36,37]. The timescales associated with this dynamic heating approach are vastly different from those experienced in conventional laser heating DAC experiments; the near-instantaneous energy deposition in the sample during single pulse irradiation produces ultrafast (sub-ns) heating followed by rapid cooling and relaxation due to heat dissipation[38]. Due to the long X-ray absorption length of $H_2O$ (~4000 μm at 40 GPa and 18 keV), the necessarily thin (<30 μm) samples were heated using various inert couplers which were directly heated by the XFEL beam, many of which are not suitable for IR laser heating due to their high reflectivity in the IR spectrum. This ability of indirect XFEL heating to induce bulk heating of $H_2O$ was recently demonstrated at PAL XFEL[38]; however, previous conclusions relied on indirect methods of melt detection due to the low repetition rate of the PAL source (30 Hz), which was insufficient to probe the hot state in situ. In our study, the structural evolution of $H_2O$ during the heating process was monitored using pulse-resolved, MHz XRD data collected using an Adaptive Gain Integrating Pixel Detector (AGIPD)[39], which enabled the tracking of structural changes, including phase transformations and chemical reactivity, with sub-microsecond temporal resolution. The development of a data analysis methodology in which diffraction spots from individual crystallites were identified in XRD images allowed us to harness the extensive volume of XRD data collected from the hot sample, which significantly improved the diffracted signal of SI ice in comparison to time-integrating methods. $H_2O$ was observed to transform from ice VII to SI-bcc on heating in the 26–69 GPa pressure range, whereas SI-fcc was only observed at pressures ≥50 GPa. The absence of SI-fcc in the lower pressure runs, despite evidence of melting and SI-bcc/fluid coexistence, is in good agreement with previous $CO_2$ laser heating studies[12,15] but contrasts with results from IR laser heating experiments in which SI-fcc was observed at pressures as low as 30 GPa[13]. The discrepancies in the reported SI-fcc phase stability fields are discussed in terms of the timescales and $P$-$T$ paths associated with different experimental approaches.

## Results and discussion
### X-ray heating
High-pressure $H_2O$ samples were indirectly heated using a variety of mid- and high-Z couplers (see "Methods" section, Supplementary Table 1, and Supplementary Fig. 1) which were irradiated with 300 XFEL pulses at a 2.2 MHz repetition rate. A number of different coupler geometries were employed, the majority of which were based on a doughnut-type crucible and two of which consisted of a dispersed nanopowder. One example of a doughnut-type design is shown in Fig. 1a-c, in which multiple couplers were embedded in a rhenium (Re) gasket and insulated from the diamonds by a thin self-insulating layer of $H_2O$. Samples were subjected to multiple heating runs at different X-ray fluences by varying the beamline transport transmission to the target, starting from a low X-ray transmission (≤1%) and increasing in steps of 0.5–10%, where a run comprises a single train of X-ray pulses striking the sample. Data from doughnut-type crucibles were collected with the XFEL beam aligned to the center of the coupler hole so that heating of the coupler was primarily performed by the tails of the focused beam (outer region of the beam spot), whereas multiple spots

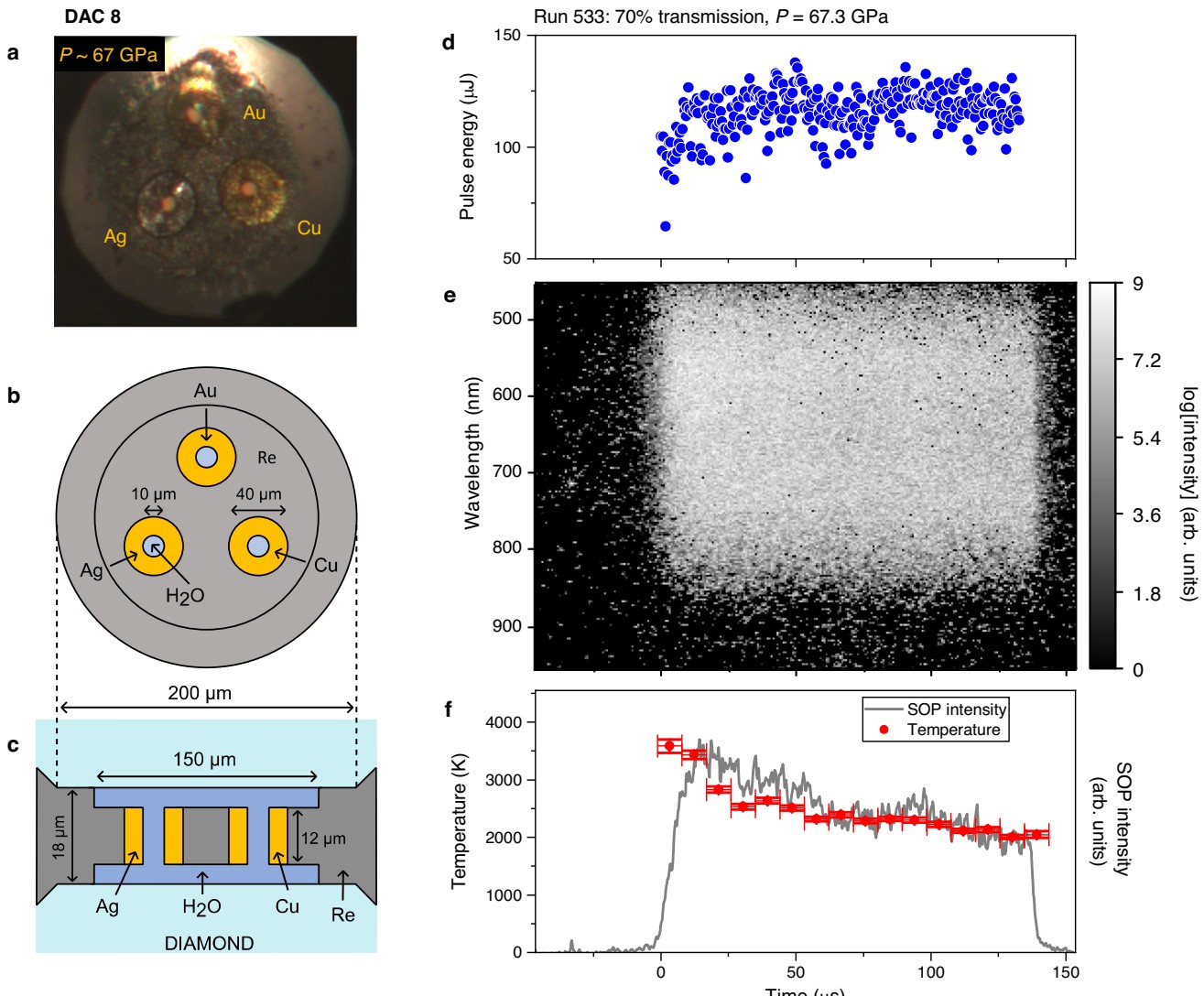

**Fig. 1 | Sample configuration and diagnostics used for X-ray heating of $H_2O$ at ~67 GPa. a** Photomicrograph showing the three doughnut-type couplers (Au, Cu, and Ag) in DAC 8, which were used to indirectly heat the $H_2O$ sample during XFEL irradiation. The couplers were imbedded in the Re gasket and insulated from the diamond anvils by a thin layer of $H_2O$, as illustrated in (**b**) and (**c**). Data were collected with the XFEL beam (<8 μm FWHM) aligned to the center of the coupler hole so that the coupler was heated by the tails of the beam. **d–f** Data from run 533, which was collected from the Cu coupler using 70% X-ray transmission. **d** Pulse energy as a function of time for the 300 pulses in the train. **e** SOP spectrogram after fluorescence removal showing the thermal emission from the coupler during the run, where time 0 corresponds to the arrival of the first XFEL pulse. **f** Temporal evolution of the total SOP intensity and temperature determined from a Planck fit to the thermal emission in a 9.07 μs time window, where the horizontal error bars indicate the bin width and the vertical error bars correspond to one half of the standard deviation confidence from the Planck fit. The temporal resolution of the SOP is not sufficient to resolve temperature oscillations during the heating/cooling process (Fig. 2a); instead, it is sensitive to the hottest part of the run where thermal emittance is the brightest. Source data for panels (**d**)–(**f**) are provided as a Source Data file.

were targeted on nanopowder couplers. Data were collected using three different X-ray focal spot sizes of <8 μm, 13 μm, and 26 μm (FWHM), which are specified in Supplementary Table 1. For some samples, the temporal dependence of the coupler temperature was constrained by streaked optical pyrometry (SOP) measurements (Fig. 1d–f).

Despite its weak scattering power, the use of intense XFEL pulses enabled the collection of high-quality XRD images from $H_2O$ using a single X-ray pulse even at the lowest X-ray transmission (0.3%). Due to the short duration of individual pulses (< 50 fs), diffraction occurs before the thermal expansion of the lattice, providing an essentially instantaneous snapshot of the sample before it is heated by the XFEL beam. XRD from the first pulse in the train therefore probes the room temperature state, and was used for pressure determination and to confirm the integrity of the sample after the previous heating run. On

longer timescales, energy transferred to the lattice via electron-phonon thermalization processes results in sub-ns heating followed by more gradual cooling via thermal conduction[38,40], where cooling rates depend on material properties (i.e. thermal conductivity) and geometry of the sample/coupler assembly. Serial pulse irradiation produces incremental heating with a saw-tooth like temporal temperature profile until a limiting state of thermal balance is achieved[40], with each XRD image providing a snapshot of the sample as it cools from the hot state created by the previous pulse. Finite Element Analysis (FEA) simulations (Fig. 2) suggest that indirectly heated materials experience reduced thermal oscillations due to the damping effect of heat diffusion; however, oscillations are still significant in the vicinity of the coupler. Overall, this dynamic heating approach reduces the total heating time to ~130 μs per run, which is ~10,000 times shorter than typical XRD exposure times at synchrotron sources[12,13,15].

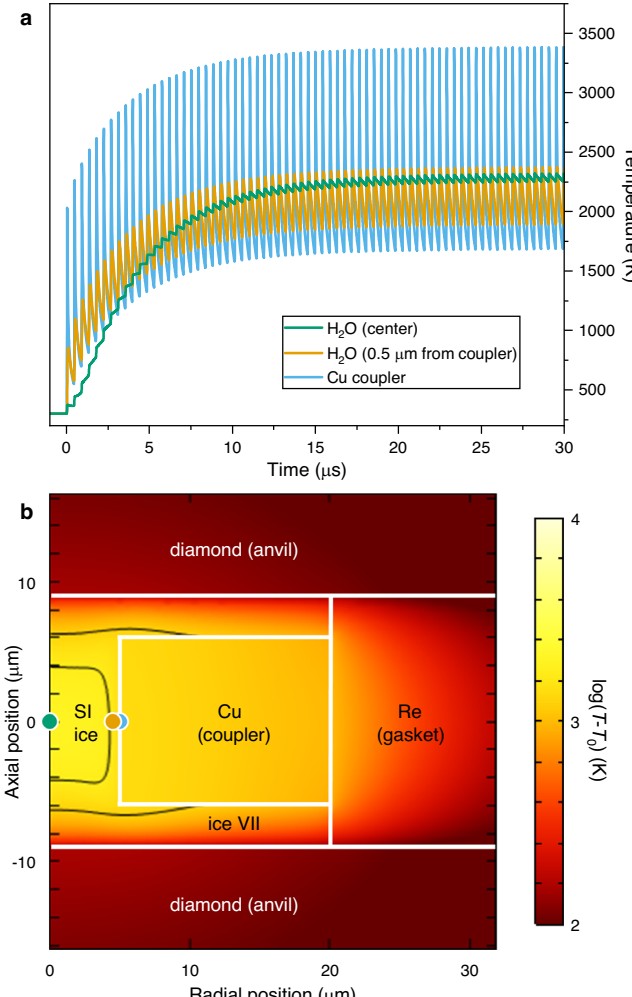

**Fig. 2 | Results of a FEA model used to simulate X-ray heating.** The model simulates X-ray irradiation of the Cu coupler in DAC 8 (run 533). **a** Temperature evolution at the coupler edge, in the crucible center, and 0.5 µm from the edge of the coupler. Although the coupler temperature undergoes continuous oscillations, these are strongly damped in the center of the crucible and the temperature stabilizes after ~20 pulses. However, temperature oscillations in the $H_2O$ are still significant in the vicinity of the Cu coupler (~500 K at 0.5 µm from the coupler edge). **b** Temperature distribution in the sample at the time of the 82nd XFEL pulse, showing conditions detected with XRD. Black lines show possible boundaries between ice VII and SI-ice, taken to be at 1500 K[12] or 2000 K[13]. The XFEL beam is incident from below. Temperatures in (**a**) are reported at the blue, green, and orange points. Source data for panel (**a**) are provided as a Source Data file.

## Observation of body-centered cubic superionic ice

Evidence of SI-bcc was observed in data collected from 9 DACs between 26 and 69 GPa (Supplementary Table 1). The evolution of XRD patterns collected during irradiation was similar for all runs (Supplementary Figs. 2 and 3), illustrated here by a dataset collected from an Ag doughnut-type coupler at 69.3 GPa (Fig. 3). In the Ag run, heating resulted in a splitting of the ice VII and Ag reflections which increased during the first ~20 pulses, with both sets of reflections shifting to lower diffraction angles due to thermal expansion. At the same time, the growth of a new reflection at ~19.2° from pulse 3 onwards is consistent with the expected *d*-spacing of the (110) SI-bcc reflection at this pressure[12,13]. No other SI-bcc reflections could be observed due to the limited 2θ detector coverage, which prevented a definitive identification of the crystal structure. However, the possibility that this peak originated from SI-fcc was ruled out because of the absence of the (200) fcc reflection, which would fall within the angular range of our

detector, and the fact that indexing this new peak as the (111) fcc reflection would imply an unphysical density almost identical to cold ice VII at the start of the run (Supplementary Table 2). The observation of SI-bcc alongside hot and cold ice VII is indicative of a significant spatial temperature gradient in the probed sample volume, which is typical in such X-ray heating experiments[41]. Although the SI-bcc (110) reflection overlaps with the weak signal from the (101) Re reflection, it is clearly identifiable in the 2D diffraction images due to the different texture of their Debye Scherrer rings. The possibility that the spots originated from the recrystallization of Re at high temperatures was ruled out because the texture of the (100) Re ring remained unchanged during the run, and because the spots identified as SI-bcc were absent in the first pulse pattern of the subsequent run.

Following the disappearance of the hot ice VII reflection around pulse 20, no significant changes were observed for the remainder of the run (Fig. 3a and Supplementary Fig. 4), suggesting that the system reached a balance between X-ray heating and heat loss via thermal conduction[40]. Based on this observation, images from pulses 51-300 were summed and integrated to produce a 1D diffraction profile from the hot sample (Fig. 4a) in which the (110) SI-bcc reflection is clearly visible alongside the broad, hot (110) ice VII reflection. Due to the spotty nature of the $H_2O$ signal, it was possible to analyze the unwrapped (2θ-φ) 2D diffraction images using a spot-finding algorithm (see Methods) to determine the 2θ position of $H_2O$ diffraction spots in each image, which was used to produce a histogram showing the total number of ice spots in images collected from pulses 51-300 (Fig. 4a and Supplementary Movie 1). The resultant histogram is in excellent agreement with the summed pattern after subtracting the Re signal, validating this approach. This analysis procedure was repeated for all runs from this sample in which SI ice was observed, and the datasets were combined to produce a single integrated profile and histogram (Fig. 4b). Although these runs were collected with different X-ray transmissions (25.5–37%) which are expected to produce different temperatures, the SI-bcc reflection is clearly distinct from that of ice VII, confirming that the observed peak separation is not an artifact relating to the large temperature gradient within the heated area. The narrow width of the SI reflection is most likely due to the narrow temperature stability field of this phase at 69 GPa, which is above the upper pressure limit of the SI-bcc stability field reported in previous work[12,13].

## Observation of face-centered cubic superionic ice

The results from the Ag coupler in DAC 8 (Figs. 3 and 4) are compared to runs collected from Au and Cu couplers in the same DAC (Supplementary Fig. 5), which were analyzed using the summed and spot-finding methods described above (Fig. 5). The SI-bcc reflection is clearly visible in both the summed XRD pattern and histogram produced from the Au runs, and the lattice parameters determined using each approach are in excellent agreement with each other ($a = 2.856$ Å and 2.854 Å from the summed pattern and histogram, respectively), and with those determined from the Ag runs ($a = 2.844$ Å and 2.843 Å) where the pressure was 5.2 GPa higher. Although the summed pattern from the Cu runs is difficult to interpret due to the broad background originating from the detector behavior at high signal levels, the SI-bcc reflection is clearly visible in the histogram alongside a second, well separated, lower-angle reflection. Indexing this peak as the (111) SI-fcc reflection determines a density of 2.618 g/cm³, which is almost identical to the SI-bcc density of 2.619 g/cm³, in agreement with previous work which found that both SI phases can be described by the same equation of state (EoS)[13]. In addition to the observation of the (111) SI-fcc reflection in the 90 and 100% transmissions runs (Fig. 6), the (200) SI-fcc reflection is also present in the 80% and 100% runs (Supplementary Fig. 6), further confirming the nature of this phase. On subsequent re-examination of the Ag data, a small number of isolated SI-fcc diffraction spots were located in several XRD images from 2 different runs (Supplementary Fig. 7), confirming that the fcc reflections

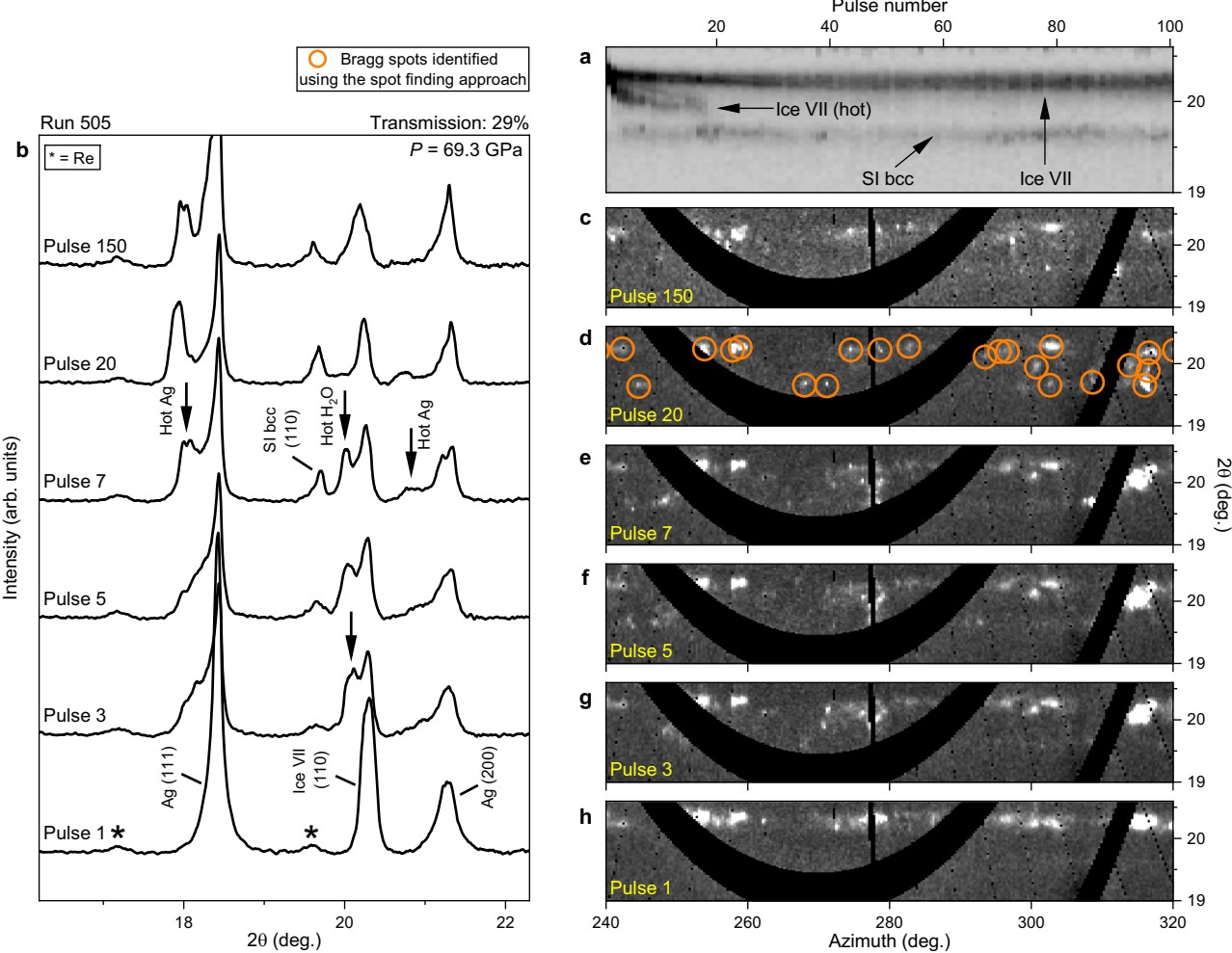

**Fig. 3 | X-ray diffraction collected during XFEL irradiation of H$_2$O at 69.3 GPa.** Data were collected using an Ag doughnut coupler (DAC 8) and an X-ray beam diameter of <8 μm FWHM. **a**, **b** Integrated XRD patterns and (**c−h**) unwrapped (2θ-φ) XRD images collected during irradiation with 300 XFEL pulses at 29% transmission (run 505). In (**a**), darker shades correspond to higher intensity. In (**c−h**), lighter shades correspond to higher intensity. Heating results in a splitting of the ice (110), Ag (111), and Ag (200) reflections, which is indicative of two distinct temperatures in the probed volume, predominantly originating from the lack of thermal insulation between the H$_2$O and the diamond anvils. Panel (**a**) illustrates how heating occurs during the first ~20 pulses, after which the diffracted signal from H$_2$O remains essentially constant. In (**d**), the diameters of the orange circles are unrelated to individual spot sizes. Source data are provided as a Source Data file.

---

in the Cu runs originated from H$_2$O rather than a chemical reaction between the sample and the coupler. However, no evidence of SI-fcc was located in any images from the Au runs from this DAC.

The difficulty in forming SI-fcc in these runs is surprising when SI-bcc formed so readily. Although SI-bcc is stable at lower temperatures than SI-fcc (Fig. 7), these runs were performed at pressures of 64-69 GPa, which are close to (or above) the previously-reported upper limit of the SI-bcc pressure stability field. In particular, Weck et al.[12] did not observed SI-bcc above 57 GPa, and although Prakapenka et al.[13] reported a single SI-bcc data point at 67 GPa and ~2000 K, there is no evidence of SI-bcc in their 2500 K diffraction pattern at 67 GPa, despite clear evidence of SI-fcc and ice VII. Although the limited azimuthal coverage of the AGIPD may account for the absence of SI-fcc in certain runs, it is unlikely to be the only contributing factor given the consistently strong signal observed from SI-bcc. Instead, the difficulty in forming SI-fcc points to an intrinsic feature of the pump-probe X-ray heating scheme and the specific coupler materials and geometry used in these runs, combined with a large energy barrier to form SI-fcc. As XRD provides a snapshot of the sample during cooling, the sample temperature is heavily dependent on heat distribution to the surrounding media, and the high thermal conductivities of Au, Ag, and Cu (318, 429, and 401 W/mK at ambient conditions, respectively[42]) limit heating of H$_2$O by fast heat transfer away from the sample. Although the diffracted signal from hot Ag (Supplementary Fig. 8) suggests that a portion of the coupler remained close to its melting point after 443 ns, the coupler reflections in the Cu and Au runs remain essentially unchanged from the cold pattern (Supplementary Fig. 9) - despite SOP determining an average coupler temperature (Supplementary Table 3) which is well above the previously-reported lower temperature limit of SI-fcc[12,13] – suggesting that the coupler cooled significantly between subsequent XFEL pulses. These observations are supported by FEA simulations (Fig. 2a) which suggest that the Cu becomes colder than the sample during the cooling process. Enhanced cooling in the real sample compared to the FEA is attributed to the significant reduction in the H$_2$O insulation layer thickness at high pressure, potentially leading to direct contact between the coupler and the diamond anvil.

**Optimal parameters for indirect X-ray heating**
The summed XRD profiles and diffraction spot histograms are shown in Fig. 5 for DACs 1 - 8. The ice VII and SI-bcc peaks are very clearly

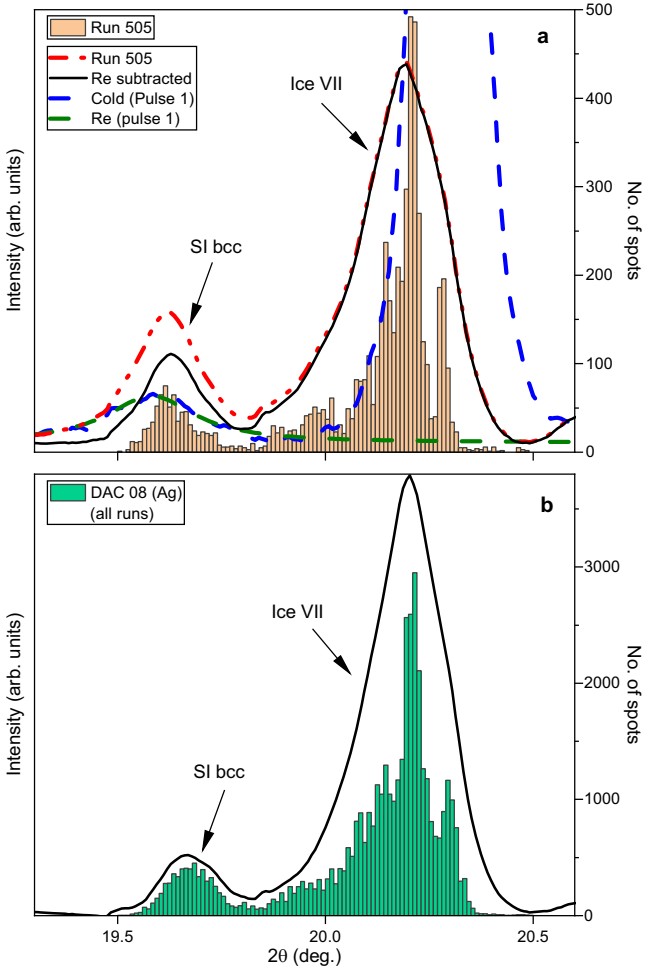

**Fig. 4 | Integrated patterns from summed images and histograms showing the results of the spot finding approach.** Data are shown for (**a**) run 505, and for (**b**) all runs collected from the Ag coupler in DAC 8 where SI was observed. The construction of the histogram for run 505 is illustrated in Supplementary Movie 1. When the Re contribution (estimated from the first pulse pattern) is subtracted from integrated patterns, they are in good agreement with the histogram. Source data are provided as a Source Data file.

### Absence of face-centered cubic superionic ice at low pressures

Examination of individual XRD images from the lower pressure (<60 GPa) runs revealed a single (111) SI-fcc crystallite (Supplementary Fig. 11) at 49.9 GPa (DAC 6). Although this does not provide conclusive evidence for the existence of SI-fcc, it is consistent with the expected stability field of this phase reported by both ref. [13],[12] and so was included in further analysis. No evidence of SI-fcc was observed in data collected from DACs 1-5 (26–38 GPa); however, the dramatic reduction in the $H_2O$ diffracted signal in the highest fluence runs collected from DAC 4 (~38 GPa) indicates that the bulk of the sample had melted. This is supported by SOP measurements, which determined the average coupler temperature to be above the ice melting temperature at this pressure. Melting is particularly striking in the 90% Ag run due to the strong signal from ice VII at the start of the run (Fig. 8 and Supplementary Movie 2), which almost completely disappears by pulse 30. Individual diffraction spots from both bcc phases reappear throughout the run, suggestive of solid-fluid coexistence. Although no evidence of diffuse scattering from the fluid was observed due to the weak scattering power of $H_2O$ and the limited azimuthal coverage of the detector, the possibility that the disappearance of the $H_2O$ solid signal originated from chemical reactivity or sample loss was ruled out by the strong ice VII signal and absence of chemical reaction products in the first pulse pattern of the subsequent run (Supplementary Fig. 12).

The absence of SI-fcc in the 38–40 GPa runs is surprising due to the strong evidence for melting and fluid/SI-bcc coexistence, and the fact that previous work[13] observed SI-fcc at pressures as low as 29 GPa. Instead, our results are in agreement with previous $CO_2$ laser heating studies[12],[15] which did not observe SI-fcc during heating at pressures up to 45 GPa, but instead observed SI-bcc to coexist with the fluid. In order to resolve the discrepancy between results obtained using different heating techniques (X-ray and $CO_2$ vs. IR laser heating), it is necessary to consider the differences and similarities between each experimental approach. The results do not appear to be correlated with the heating method (i.e. direct or indirect heating), as both Prakapenka et al. [13] and this work performed indirect heating using embedded absorbers and obtained different results. Heating durations of ≥5 s in the $CO_2$ and IR heating experiments are 4 orders of magnitude longer than XFEL heating (~130 μs), which seemingly suggests that timescale is not a factor. However, as the output power of $CO_2$ lasers is typically controlled by pulse width modulation which produces fluctuations in laser intensity[31], the time that the sample lies within SI-fcc stability field could be reduced by the resultant thermal fluctuations which could potentially cause the sample to oscillate across the SI-fcc phase boundary. Although SI-fcc may be thermodynamically stable below the melting line, previous theoretical studies calculated a lower solid/fluid interfacial free energy for the SI-bcc lattice in comparison to SI-fcc[43], suggesting that the fcc lattice may have a longer nucleation time than SI-bcc when crystallizing from the fluid. We therefore propose that the absence of SI-fcc in the 38–40 GPa XFEL heating runs is due to short heating timescales and the *P-T* path associated with XFEL pump-probe experiments, combined with kinetic hindrance associated with the formation of SI-fcc from the melt. This hypothesis is consistent with the results of IR laser heating experiments using the same approach as Prakapenka et al. (Supplementary Fig. 13), which were performed at the GSECARs beamline at the Advanced Photon Source using Au flakes as a laser absorber. After initially heating $H_2O$ above its melting temperature, molten ice was first observed to coexist with SI-bcc when the laser power was reduced, whereas the observation of SI-fcc on further decrease of the laser power coincided with the disappearance of the diffuse scattering signal. The observation of liquid/SI-bcc coexistence is attributed to localized heating and melting of the $H_2O$ in the vicinity of the Au flakes, which drives the movement of the Au particles and results in rapid melting and recrystallization of $H_2O$. Consequently, the sample does not remain in the SI-fcc stability field long enough for SI-fcc to form. Similarly, thermal fluctuations in the vicinity of the coupler

separated at all pressures, particularly in the histogram plots, due to the lower photon energy (18 keV) compared to previous DAC experiments (>30 keV). Bulk heating is particularly pronounced in runs collected from the Rh doughnut-type coupler in DAC 9 (Supplementary Fig. 10), evidenced by the strong SI-bcc signal and almost complete disappearance of ice VII signal, most likely due to a combination of optimal geometrical parameters and coupler properties. In particular, the lower thermal conductivity of Rh (150 W/mK at ambient conditions) compared to Au, Ag, and Cu is expected to reduce heat dissipation in the unheated coupler, resulting in a more uniform temperature distribution in the sample. However, although Rh was the most promising coupler in terms of heating efficiency, the appearance of a number of unidentified diffraction spots after the first observation of SI-bcc in some runs suggests that Rh reacts with SI $H_2O$. Data from DAC 9 were therefore not analyzed using the spot finding algorithm, but are presented to illustrate that uniform heating of low-Z samples using a high-Z coupler is possible by careful choice of X-ray parameters, sample geometry, and material parameters. Although chemical reactivity was a general problem for Rh couplers, one run from DAC 7 was included in the analysis because reaction products were not observed until the end of the run, after all traces of SI-bcc had disappeared.

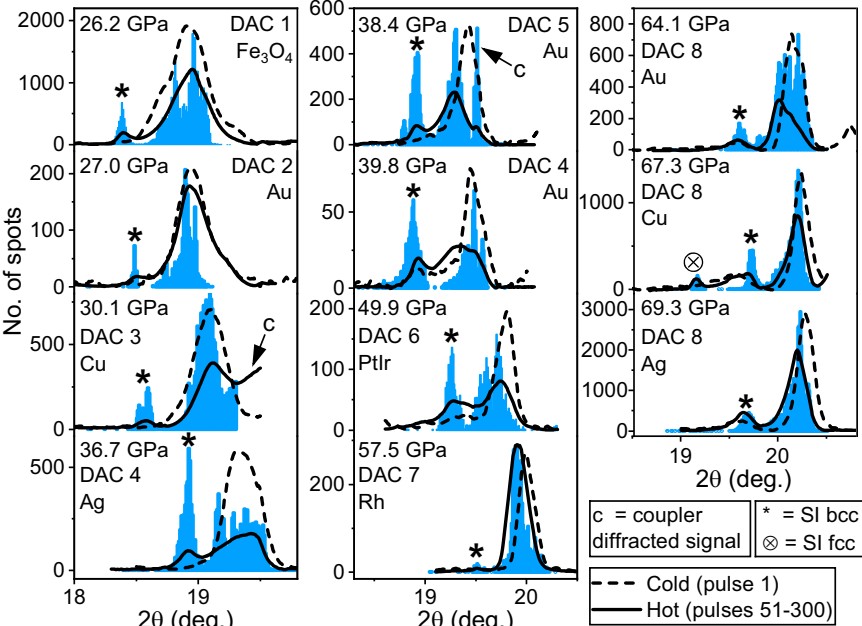

**Fig. 5 | Histograms showing the number of H₂O diffraction spots in runs collected from different samples.** For each sample, histograms were compiled from XRD data collected from pulses 51-300 in runs where SI ice was observed. The width of each bin is 0.01 degrees. Histograms are compared with integrated XRD patterns produced from the summed Adaptive Gain Integrated Pixel Detector images of the hot (pulses 51–300) and cold (pulse 1) sample. The *y*-axis refers to the histogram plot, and the integrated patterns from the hot and cold sample were normalized to a single arbitrary scaling factor for the figure. The coupler material is specified in each case. The high-angle signal in the DAC 3 and 5 runs are from the hot coupler. With the exception of DAC 1 and DAC 2, in which the coupler was in the form of a dispersed nanopowder, all couplers were of a doughnut-type design. Source data are provided as a Source Data file.

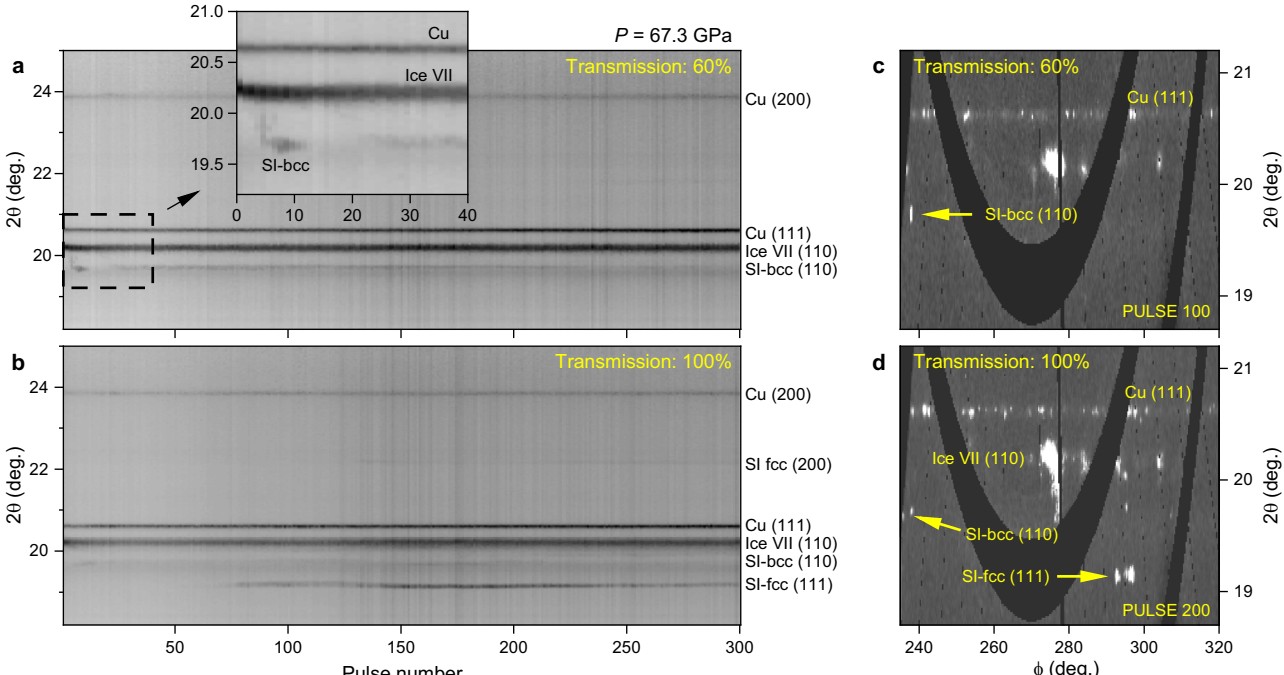

**Fig. 6 | XRD data collected during X-ray heating of H₂O at 67.3 GPa.** Data were collected during irradiation of the embedded Cu doughnut coupler in DAC 8 using an X-ray beam size of <8 μm FWHM. Integrated XRD data are shown from runs (**a**) 532 and (**b**) 536, and single pulse, unwrapped (2θ-φ) XRD images from runs (**c**) 532 and (**d**) 536 are shown for a reduced 2θ range. In (**a–b**), darker shades correspond to higher intensity, whereas in (**c–d**), lighter shades correspond to higher intensity. The data shown in (**a**) and (**c**) were collected using 60% transmission, and the (110) SI-bcc reflection is observed from pulse ~5 onwards (see insert), The SI-bcc diffracted signal is predominantly from several intense spots, such as the one highlighted in (**c**). The data in (**b**) and (**d**) were collected at 100% transmission, and both SI phases (bcc and fcc) are observed. Source data are provided as a Source Data file.

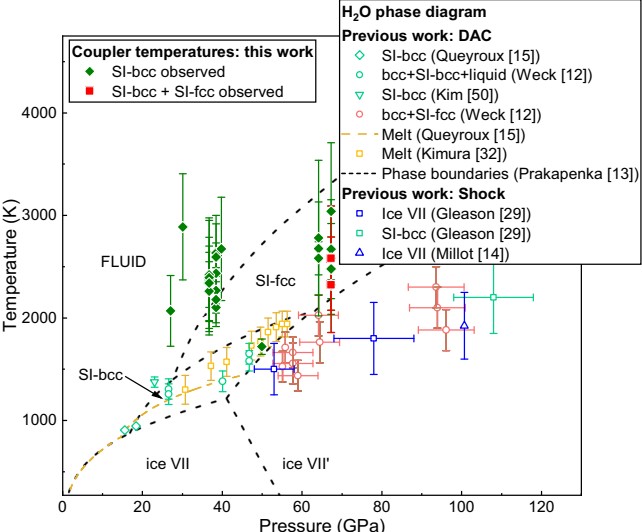

**Fig. 7 | Average coupler temperatures in experiments where SI ice was observed.** Data points correspond to the average SOP temperature during the run, and the error bars show the standard deviation. Source data are provided as a Source Data file. Thermal emission was not detectable for all samples, and data points are shown for runs in which thermal emission was observed. Due to the low emissivity of $H_2O$, SOP measures thermal emission from the coupler surface, which does not necessarily correspond to the temperature in the $H_2O$ sample. Our data points are compared to the phase diagram of $H_2O$ reported in previous studies. The text labels indicate the phase stability regions reported in ref. 13. For simplicity, we do not make any distinction between ice VII and VII′ in the main text. SI-bcc and SI-fcc data points from previous static[12,15,50] and shock compression[14,29] studies are included for comparison, as well as ice VII data points from shock compression work[14,29] and the $H_2O$ melting curves from ref. 15,32. Ice VII data points from static compression studies are not included to avoid overcomplicating the figure. The data points from ref. 15. indicate the SI-bcc/ice VII phase line determined on isothermal compression.

in the XFEL heating experiments (Fig. 2a) due to the pulsed nature of the XFEL beam can account for fast recrystallization of SI-bcc from the fluid in the 90% transmission run collected from DAC 4 (Fig. 8), suggesting an approximate lower bound of 443 ns for the nucleation time. The observation of SI-fcc at higher pressures in the XFEL experiment is attributed to the higher melting temperature of $H_2O$ at this pressure and the fact that the average coupler temperature lies in the stability field of SI-fcc, which avoids transforming a portion of the sample into the fluid phase. This is consistent with the observation of large SI-fcc crystallites at the same azimuthal position for multiple frames in the 100% transmission run collected from DAC 8 (Fig. 6), rather than small, recrystallizing spots which are typically formed during fast quenching. The possibility of timescale- and path-dependent transitions in high-temperature $H_2O$ should therefore be taken into account when comparing results obtained using different experimental techniques; in particular, considering the conflicting results of recent shock compression experiments (Millot et al. observed SI-fcc near 160 GPa and 3200 K[14], whereas Gleason et al.[29] saw SI-bcc as pressures as high as 207(10) GPa at 5500(500) K).

## Density of superionic ice

The histograms in Fig. 5 were used to determine the densities of both SI phases, which are compared with the results of previous studies in Fig. 9. Our SI-bcc points are in good agreement with the results previous studies[12,13]; our highest-pressure data points lie closer to the Prakapenka et al. [13] EoS which includes a thermal pressure correction, despite the fact that such a correction was not included in our analysis, whereas our lower-pressure points lie closer to their uncorrected curve. The SI-fcc densities determined in this study are in excellent agreement with those of SI-bcc at the same pressure, in agreement with Prakapenka et al. [13], but lie at lower densities than those reported by Weck et al. [12] at the SI-bcc/SI-fcc transition temperature. Weck et al.[12] reported the density of SI-fcc to be strongly temperature dependent, and our SI-fcc data are in better agreement with the density they report at 57 GPa and 1927 K, which is ~500 K above their reported SI-bcc/SI-fcc

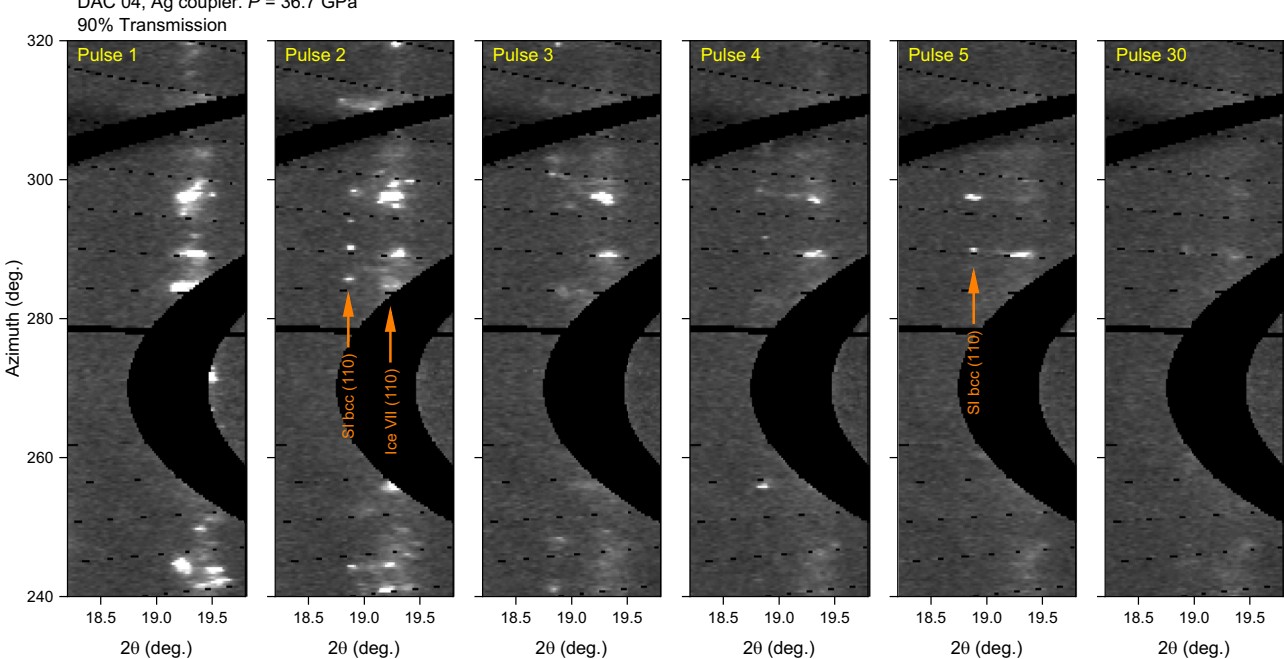

**Fig. 8 | Unwrapped (2θ-φ) XRD images showing evidence of melting of $H_2O$.** Data were collected using 90% X-ray transmission from the embedded Ag coupler in DAC 4 (36.7 GPa) using an X-ray beam size of <8 μm FWHM (run 917:17). Lighter shades correspond to higher intensity. Source data are provided as a Source Data file. The significant reduction in the intensity of the ice VII/SI-bcc diffracted signal during the run is attributed to melting of $H_2O$. No evidence of the SI fcc (111) reflection was observed, which would be expected to be present at approximately 18.4°. Unwrapped images from the full run are shown in Supplementary Movie 2.

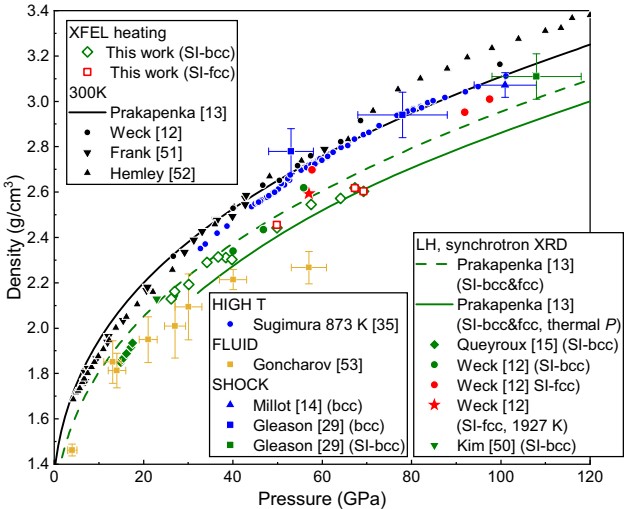

**Fig. 9 | Density as a function of pressure for different phases of H₂O.** The data points from this work show the SI-bcc and SI-fcc densities calculated from the histograms in Fig. 5. Source data are provided as a Source Data file. The data point from DAC 9 (34.2 GPa) is included to illustrate the agreement with other data, despite weak evidence of chemical reactivity in this sample. Our results are compared with SI ice data from previous DAC experiments[12,13,15], which are identified as using laser heating (LH-DAC) or resistive heating (RH-DAC) techniques. Ice VII data from ambient temperature[12,13,52] and high temperature[35] DAC experiments, shock compression experiments[14,29], as well as data from the high temperature fluid[53], are shown for comparison. In addition to the data points taken from the main paper of Weck et al.[12], which indicate the SI-bcc and SI-fcc densities at the transition temperature, their SI-fcc data point at 57 GPa and 1927 K is also included to indicate the volume of thermally-expanded SI-fcc at higher temperature.

transition temperature. Although they report a corresponding SI-bcc density for the 57 GPa run which is significantly larger (~2.67 g/cm³)[33] than our measured SI-bcc density at the same pressure (~2.55 g/cm³), our data are in good agreement with the SI-bcc density determined from the 1927 K XRD pattern in Fig. S4 of their supplementary material (~2.55 g/cm³), suggesting that the reported SI-bcc density actually corresponds to that of expanded ice VII. The possibility that SI-fcc was not observed in our XFEL experiments until ~500 K above the transition temperature is intriguing considering the difficulty in forming SI-fcc, and is consistent with a large energy barrier associated its formation. However, due to the lack of accurate temperature measurements in our XFEL experiments, it is not possible for us to resolve discrepancies between previous work. Overall, the large density difference between ice VII and SI-bcc reported in this work contrasts with the 108 GPa SI-bcc data point from the shock compression study of Gleason et al.[29], which suggests that the two phases have similar densities. However, as their SI-bcc data point lies outside the SI stability field reported by ref. 13, it is possible that that this sample was actually in the ice VII stability field.

In conclusion, an indirect XFEL heating technique was used to investigate the structural properties of SI ice, demonstrating that stepwise XFEL heating is a viable method to study low-Z materials such as H₂O at high pressures and temperatures. Evidence of SI-bcc was identified in a large number of runs performed at 27-69 GPa, illustrating that this approach is able to detect isostructural transitions, and SI-fcc was observed in runs ≥50 GPa. The large volume of XRD data collected using a pulse-resolved, MHz detector, combined with the highly textured nature of the SI-bcc diffraction signal, motivated the development of a new data analysis approach in which the identification of diffraction from individual crystallites in XRD images was used to enhance the signal from high-temperature SI phases and determine their density as a function of pressure. No evidence of SI-fcc was

observed in lower pressure runs, despite clear evidence of melting and the simultaneous observation of SI-bcc and fluid at ~40 GPa. Based on previous theoretical studies which calculated a lower solid/fluid interfacial free energy for the SI-bcc lattice in comparison to SI-fcc[43], and considering prior observations of SI-fcc forming at these *P-T* conditions upon heating[13], we attribute the absence of SI-fcc in these experiments to the short heating timescale combined with the *P-T* path of the pump-probe approach in which SI ice is formed on cooling from the melt. The results may have important implications for the stability of SI phases in solar and extrasolar ice-rich planets during dynamical freezing such as in internal convection processes, where the preferential crystallization of SI-bcc from the fluid may result in different physical properties of the solid ice (e.g. electrical and thermal conductivity) across the same internal layer that in turn, may affect interior dynamics and magnetic fields.

## Methods
### Sample preparation
H₂O samples were loaded into a total of 9 symmetric DACs (DACs 1–9) with standard design diamonds mounted on the upstream side and Boehler Almax diamonds facing downstream. Microscope images of the samples before the XFEL experiment are shown in Fig. 1a and Supplementary Fig. 1, which were collected after the sample was compressed to the final pressure for data collection. A total of 5 different coupler materials were employed (Au, Ag, Cu, Rh, and Fe₃O₄) in different coupler geometries (nanopowder, doughnut, and embedded doughnut) which were designed with the aim of minimizing the temperature gradient in the indirectly heated H₂O samples (for preparation details see Supplementary Note 1). With the exception of DAC 7, all samples were loaded without a thermal insulation layer, relying instead on a self-insulating H₂O layer close to the diamonds to act as a thermal barrier. This avoided the need for commonly-used insulation layers such as Al₂O₃, LiF etc., which can complicate the interpretation of diffraction patterns and/or result in unwanted chemical reaction products. After loading, samples were pre-compressed to the desired pressure for the XFEL experiment

### Experimental details: sample screening at PETRA III
After compression to the desired pressure for the European XFEL experiment, samples were screened at the P02.2 beamline at the PETRA III synchrotron source to confirm the integrity of the sample loading using XRD. The screening was performed using 25.6 keV X-rays focused to 3(v) × 8(h) μm² (FWHM) using a series of compound refractive lenses (CRLs), and XRD data were collected using an XRD1621 area detector (PerkinElmer). After the XFEL experiment, a 2D diffraction map collected from DAC 8 found no evidence of chemical contamination. This was performed at the P02.2 beamline using 42.7 keV X-rays focused to 2(v) x 2(h) μm² (FWHM) using Kirkpatrick–Baez mirrors, and data were collected using an XRD1621 area detector.

### Experimental details: European XFEL parameters
X-ray heating experiments were performed at the High Energy Density (HED) Instrument[44] at the European XFEL using the dedicated DAC platform in Interaction Chamber 2[36]. The experiment was performed using an X-ray photon energy of 18.047 keV and a 2.2 MHz intra-train repetition rate, which corresponds to a 443 ns spacing between consecutive pulses, where the photon energy was calibrated using the HIgh REsolution hard X-ray single-shot (HIREX) spectrometer[45]. X-rays were focused using a series of CRLs using three different configurations to achieve focal spot sizes of <8 μm, 13 μm, and 26 μm (FWHM), where the beam size was estimated from edge scans using a polished W rod. Accurate determination of the beam diameter for the smallest focal spot size was not possible based on the edge scan data due to erosion of the W rod.

## Experimental details: X-ray diffraction at European XFEL

Pulse-resolved XRD data were collected using a 500k AGIPD detector positioned outside of the vacuum chamber at a sample-to-detector distance (SDD) of ~430 mm, which provided an angular coverage of ~7.5 to 27 degrees ($q = 4.27$-$1.20$ Å$^{-1}$). The detector geometry (SDD, tilt, and rotation) was calibrated using a $Cr_2O_3$ diffraction standard (NIST SRM 674b) using the Dioptas software[46]. Radial integration of diffraction images was performed using Dioptas and azimuthally unwrapped ($2\theta$-$\varphi$) diffraction images were produced using FIT2D[47]. Prior to integration, the intensity of individual diffraction images was scaled using the pulse-resolved intensity and position monitor (IPM)[36] positioned upstream of the sample to account for fluctuations in the pulse energy across the pulse train, and all images within the run were normalized to the mean IPM value within the run to allow for comparison of runs collected at different levels of X-ray attenuation. The X-ray fluence incident on the sample during the run was controlled using a series of solid attenuators, which enabled the X-ray transmission to be varied from 0.3 to 100%. The pulse energy on target was determined using the IPM, which was calibrated using an X-ray gas monitor (XGM)[36] at the start of the experiment.

Alignment of the sample to the X-ray beam was performed based on visual observation using an optical imaging system. This alignment method resulted in an offset in the sample position along the X-ray beam direction with respect to the diffraction calibrant due to refraction from the downstream diamond anvil, which was corrected by changing the SDD used in the detector calibration based on the measured diamond thickness and refractive index. The X-ray position on the optical camera was determined by exposing an area on the gasket for several seconds at low fluence, which produced a small dark region resulting from X-ray damage. This approach was found to be sufficient to align the XFEL beam with small holes (6–22 μm diameter) in doughnut couplers, where the XFEL beam was positioned at the center of the hole to maximize the diffracted signal from the $H_2O$ sample. However, the positional jitter in the XFEL beam position (approximately equal to the X-ray focal spot size) meant that the beam position varied from run to run, which could potentially result in different degrees of X-ray heating in subsequent runs due to the large difference in absorption lengths of $H_2O$ and the mid/high-Z couplers.

## Experimental details: streaked optical pyrometry

The temporal evolution of the temperature profile during the heating run was determined from streaked optical pyrometry (SOP) measurements performed using a streak camera (C13410-01A, Hamamatsu) coupled to an optical spectrometer (IsoPlane 160, Princeton Instruments), which collected thermal emission from a 50 μm diameter region on the upstream side of the sample. Full details of the SOP system are given in ref. 49. In all cases, data were collected using a 200 μs streak window to cover the entire duration of the 132.9 μs long X-ray pulse train. SOP data collected using a low X-ray fluence were dominated by a fluorescence signal originating from the diamond anvils and/or $H_2O$ media, which was most prominent in the short wavelength (~500 nm) range. The presence of thermal emission at higher X-ray fluence was identified by a discontinuous increase in the total SOP intensity as a function of incident pulse energy when all runs from the same coupler were compared. For runs in which thermal emission was observed, the fluorescence signal was removed by subtracting the fluorescence-only spectrogram collected at the highest X-ray transmission. After the fluorescence correction was applied, the temperature was determined by fitting a Planck function to the 600–775 nm wavelength region of the thermal emission spectrum from each 9.1 μs temporal window assuming gray-body emission. The reliability of each temperature measurement was evaluated based on the criteria outlined in ref. 49. Examples of SOP data treatment, including the fluorescence correction, are shown in

Supplementary Fig. 14. A summary of SOP temperatures from all runs are given in Supplementary Table 3, and correspond to the average over the entire run. Due to the low emissivity of $H_2O$ in the temperature region investigated in this work, SOP provided a measurement of the coupler surface temperature at the hot spot, rather than a direct measurement of the sample temperature, which is expected to be lower than that of the coupler (Fig. 2a). As the total radiated energy is proportional to $T^4$, the strongest contribution to the thermal radiation spectrum is from the hottest region on the coupler surface. Thermal emission was not detectable for all samples, which we attribute to variations in the volume of heated coupler material due to a combination of factors such as X-ray focal spot size and coupler geometry.

## Experimental details: IR laser heating at GSECARS

IR laser heating experiments were performed at the GSECARS undulator beamline (sector 13, APS, ANL). The experiment was performed using 37.07 keV X-rays focused to $3 \times 4$ μm$^2$ (FWHM), and XRD data were collected using a MAR-165 CCD (charge-coupled device). Doubled-sided, coaxial heating of samples was performed using a near-IR (1064 nm) laser with a 10 μm diameter flat-top focal spot[49]. The temperature was determined using spectroradiography measurements performed using a Princeton grating spectrometer (300 mm focal length) combined with PIXIS and PI-MAX3 CCD array detectors, and thermal emission was collected from both sides of the heated sample. For this experiment, the $H_2O$ sample was thermally insulated from the diamond by a layer of $SiO_2$, and small flakes of Au were used as the laser absorber.

## Data analysis: X-ray diffraction

Out of a total of 400 runs collected from $H_2O$ samples, evidence of SI-bcc was identified in 68 runs (Table SI) which were collected using a range of different coupler materials, geometries, and X-ray transmissions. In all cases, the (110) SI-bcc reflection was clearly distinct from the broad, hot ice VII reflection, consistent with the expected behavior for an isostructural transition. Two different approaches were used to evaluate the XRD patterns collected from the hot sample. In the first approach, images collected from pulse 51–300 from each run contained SI were summed to produce a single integrated XRD profile for each sample. For DACs in which multiple couplers were used (DACs 4 and 8), individual diffraction profiles were produced for each coupler type. For each sample, the cold pattern was produced by summing the first pulse XRD images from the runs used for data analysis, which determined the average pressure across all runs. In order to avoid systematic uncertainties introduced by the choice of EoS for different coupler types, the sample pressure was estimated based on the position of the (110) ice VII reflection in the cold pattern using the EoS of Prakapenka et al.[13]. The result of the summed approach is illustrated for DAC 1 in Supplementary Fig. 15.

For construction of the histograms shown in Fig. 5, a spot finding algorithm was used to identify the $2\theta$-$\varphi$ positions of individual diffraction spots in the unwrapped ($2\theta$-$\varphi$) diffraction images using a custom python code. First, a threshold of $\bar{i}_{bk} + n\sigma_{i_{bk}}$ was applied, where $\bar{i}_{bk}$ and $\sigma_{i_{bk}}$ are the mean and standard deviation, respectively, of the background intensity of a region on the detector in which no diffraction lines were observed. A value of $n = 6$ was used in all cases except for the Cu coupler in DAC 8, where $n = 5$ was used due to an increase in detector noise at the highest X-ray fluence due to issues related to gain switching. Individual reflections were identified as connected regions in the threshold image, and those with a pixel size of <2 were assumed to due to detector noise and discarded. The number of local maxima was computed for each of the identified regions. If multiple local maxima were present in a single identified region and separated by at least 3 pixels, the region was split into

multiple reflections using a watershed algorithm. Finally, the angular coordinates (2θ-φ) of each reflection was determined from its center of mass, and the resultant list of 2θ positions was used to produce the histogram. The value of $n$ was chosen by visual inspection of the number of spots located in the image (Supplementary Fig. 16). However, although the choice of n determined the number of spots which were identified, histograms produced using different values of $n$ were found to be in good agreement with one another (Supplementary Fig. 17).

## Finite Element Analysis

Finite Element Analysis (FEA) of sample heating, accounting for sample geometry, the materials used and their thermochemical parameters at relevant high pressure conditions, were performed using previously described techniques[38,40,41,48]. FEA was used primarily in planning experiments, to establish beam properties and sample designs to avoid coupler melting and damage, achieve sufficiently high temperature in the water sample, optimize heat transfer to the water, and minimize temperature gradients in the crucible. While major predictions of the models were confirmed by pyrometry measurements during experiments, including absolute temperatures reached and the achievement of a quasi-steady state at longer timescales, models predict a gradual rise in temperature at early times which contrasted with an observed decrease with time from elevated initial temperature. This effect could be related to beam misalignment with holes, coupler melting and movement as well as changes in sample optical properties at high temperature[41]. Due to the use of longer sweep windows with lower time resolution in SOP, the individual heating and cooling events seen in the models are not resolved[48]. For the model in Fig. 2, representative high-pressure parameters of thermal conductivity $k$, heat capacity at constant pressure $C_P$, density $\rho$, and absorptivity $\alpha$ were used for the $H_2O$ sample ($k = 20\,\mathrm{W\,m^{-1}\,K^{-1}}$, $C_P = 3050\,\mathrm{J\,kg^{-1}\,K^{-1}}$, $\rho = 2694\,\mathrm{kg\,m^{-3}}$, $\alpha = 201\,\mathrm{m^{-1}}$) and the Cu coupler ($k = 450\,\mathrm{W\,m^{-1}\,K^{-1}}$, $C_P = 472\,\mathrm{J\,kg^{-1}\,K^{-1}}$, $\rho = 9500\,\mathrm{kg\,m^{-3}}$, $\alpha = 4.41 \times 10^4\,\mathrm{m^{-1}}$), along with standard parameters for the gasket and diamonds[40].

## Data availability

The DOI for the original European XFEL data is: https://doi.org/10.22003/XFEL.EU-DATA-002590-00, and will be publicly available after the embargo period of 3 years. Source data are provided with this paper.

## Code availability

The computer code used to generate the results reported in this study is available from the corresponding author upon request.

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

## Acknowledgements

We acknowledge European XFEL in Schenefeld, Germany, for provision of X-ray free-electron laser beam time at Scientific Instrument HED (High Energy Density Science) and would like to thank the staff for their assistance. These data were collected as part of a DAC community proposal (#2590, by McMahon and Husband: https://doi.org/10.22003/XFEL.EU-DATA-002590-00). The authors are indebted to the HIBEF user consortium for the provision of instrumentation and staff that enabled this experiment. We acknowledge DESY (Hamburg, Germany), a member of the Helmholtz Association HGF, for the provision of experimental facilities. Parts of this research were carried out at PETRA III (beamline P02.2). V.B.P. and S.C. acknowledge the support of GeoSoilEnviroCARS and National Science Foundation – Earth Sciences (EAR - 1634415). We acknowledge E. Shevchenko (CNM, ANL), who synthesized the nano-Fe₃O₄ samples. J.D.M. acknowledges support from AWE CASE studentship P030463429. Support is acknowledged from the U.K. Engineering and Physical Sciences Research Council (EPSRC) Grant Nos. EP/R02927X/1 (E.J.P. and M.I.M.) and EP/P024513/1 (R.S.M). M.F. Acknowledges DOE FES funding FWP100182. Y.L. is grateful for support from the Leader Researcher programme (NRF-2018R1A3B1052042) of the Korean Ministry of Science and ICT (MSIT). This work was performed under the auspices of the U.S. Department of Energy by Lawrence Livermore National Laboratory under Contract No. DE-AC52-07NA27344 (H.C.). This research was supported through the European Union's Horizon 2020 research and innovation Programme (ERC grant 864877, H.M., and 101002868, R.S.M.) as well as UKRI STFC grant ST/V000527/1 (H.M.). A.F.G. and E.E. are grateful for the support of Carnegie Science and NSF EAR-2049127. A.F.G. is grateful for the support of NSF CHE- 2302437. S.N. and L.A. acknowledge financial support from Sorbonne University under grant Emergence HP-XFEL. We acknowledge support from the Deutsche Forschungsgemeinschaft (DFG) Research Unit FOR 2440 grants SA2585/5-1 (R.J.H, A.M., C.S.V., and H.P.L) and AP262/2-2 (K.A.).

## Author contributions

R.J.H., M.I.M., H.P.L., J.D.M., R.S.M., A.F.G., V.B.P., Z.K., C.S., C.S.V., M.F., K.A., O.B.B., R.B., A.L.C., H.C., Y.L., H.M., S.N., E.P., C.Pe., C.Pr, and U.Z. were involved in the conception of the experiment and writing of the proposal. R.J.H., M.I.M., H.P.L., J.D.M., R.S.M., A.F.G., V.B.P., E.E., S.C., Z.K., C.S., C.S.V., M.F., L.A., K.A., O.B.B., R.B., J.B., V.C., J.C., A.L.C., H.C., H.H., E.K., Y.L., X.L., H.M., A.M., M.N., S.N., E.P., C.Pe., C.Pr, and U.Z. participated in discussions of the experimental approach and data analysis. R.J.H., J.D.M., V.P., S.C., E.E., and A.F.G. prepared the samples. R.J.H. performed the data analysis and wrote the manuscript with input from all authors. R.S.M. performed the FEA calculations. R.J.H, H.P.L., J.D.M, C.S.V., L.A., O.B.B. A.M., S.N., C.Pr., and M.I.M. performed the experiment at European XFEL, and R.S.M, A.F.G., V.B.P., E.E., S.C., R.B., J.B., J.C., A.L.C., H.C., H.H., E.K., Y.L., X.L., H.M., E.P., and C.Pe. provided remote data analysis support. Z.K., C.S., K.A., V.C., C.B., A.D., and M.N. provided support at the HED instrument (European XFEL). H.G., T.L., S.S., and J.S.-D. provided support for the AGIPD detector. R.J.H., J.D.M., and H.H. performed screening experiments P02.2 beamline (PETRA III), and H.P.L. provided support at P02.2.

## Funding

## Competing interests

The authors declare no competing interests.

## Additional information

[1]Deutsches Elektronen-Synchrotron DESY, Hamburg, Germany. [2]SUPA, School of Physics and Astronomy, and Centre for Science at Extreme Conditions, The University of Edinburgh, Edinburgh, UK. [3]Carnegie Science, Earth and Planets Laboratory, Washington, DC, USA. [4]The University of Chicago, Center for Advanced Radiation Sources, Chicago, IL, USA. [5]European XFEL, Schenefeld, Germany. [6]Universität Münster, Institut für Mineralogie, Corrensstraße 24, Münster, Germany. [7]SLAC National Accelerator Laboratory, California, USA. [8]Institut de Minéralogie, de Physique des Matériaux et de Cosmochimie (IMPMC), Sorbonne Université, Paris, France. [9]Institute of Radiation Physics, Helmholtz-Zentrum Dresden-Rossendorf, Bautzner Landstraße 400, Dresden, Germany. [10]Lawrence Livermore National Laboratory, Livermore, CA, USA. [11]Department of Earth Sciences, University of Oxford, Oxford, UK. [12]Department of Earth System Sciences, Yonsei University, Seoul, Korea. [13]CEA, DAM, DIF, 91297 Arpajon, France; Université Paris-Saclay, CEA, Laboratoire Matière en Conditions Extrêmes, Bruyères-le-Châtel, France. [14]Institute of Earth and Environmental Sciences, University of Freiburg, Freiburg, Germany. [15]Present address: Bayerisches Geoinstitut, Universität Bayreuth, Universitätsstraße 30, Bayreuth, Germany. [16]Present address: Department of Earth and Environmental Sciences (DISAT), University of Milano-Bicocca, Milan, Italy. [17]Present address: School of Earth Sciences and Environmental Engineering, Gwangju Institute of Science and Technology, Gwangju, Republic of Korea. [18]Present address: Diamond Light Source Ltd, Harwell Science and Innovation Campus, Didcot, Oxfordshire, UK. [19]Present address: Synergetic Extreme Condition High-Pressure Science Center, State Key Laboratory of Superhard Materials, College of Physics, Jilin University, Changchun, China. [20]Present address: X-Spectrum GmbH, Luruper Hauptstraße 1, Hamburg, Germany.
✉e-mail: rachel.husband@desy.de; M.I.McMahon@ed.ac.uk

