## [Peer Review File · Nature Communications]

Phase transition kinetics of superionic H₂O ice phases revealed by Megahertz X-ray free-electron laser-heating experimentsREVIEWERS' COMMENTS

Reviewer #1 (Remarks to the Author):

This paper presents a detailed study of the structure and temperature of ultrafast heating of statically compressed water. They present a new method for heating using the XFEL itself. The same XFEL then probes the heated sample to create a 300 pulse record of the evolution of the sample during equilibration and cooling. The authors motivate the work with astrophysical implications- but the connection is weak and I'm not sure the timescales here are relevant. They find that SI-bcc ice is more readily nucleated from the fluid than the competitive SI-fcc phase- this is the main finding in the paper. The quality of the work is excellent and thorough.

However, I don't think that the paper is well suited for Nature Communications as it is most suited for a methodology journal. This is because the paper focuses on the details of the pump-probe X-ray heating scheme including the specific coupler materials and geometry used in this extensive study. Because they do not have accurate temperature measurements to explain discrepancies between previous work and the focus on the methodology, I don't think the scientific results reported here are suited for Nature Communications. But, I do think these results are important to publish elsewhere for future experiments using this heating techniques at XFELs.

Reviewer #2 (Remarks to the Author):

The paper by R. Husband et al. describes a high-pressure study on superionic ice formed by indirect heating through consecutive X-ray free electron laser pulses. Superionic ice was shown in literature to form at pressures above 20 GPa through different pathways, using e.g. shock compression or optical laser-heating. Two crystal structures of the oxygen sublattice have been found, namely bcc and fcc. The location of the boundary between the two states remained unclear due to different findings. In contrast to previous studies, the current paper is using an interesting new method, where a donut shaped metal coupler is heated through a fs X-ray pulse, simultaneously the ice inside the donut is probed. 300 consecutive pulses continuously increase the temperature of the metal, while the water undergoes a heating cooling oscillation, with overall temperature increase.

The paper is well written and clearly shows the appearance of the bcc structure at pressures below 50 GPa, while fcc is observed only above that threshold. The results are carefully analyzed and critically discussed. The presence of fcc exclusively at higher pressures, supports earlier findings, and resembles an important result, helping to better understand water's phase diagram and in particular the region of superionic ice above ice VII.

I recommend publication of the paper.

Few minor comments:

- In the results and discussion section (page 6) and/or Figure 1 the X-ray beams size should be clearly provided. Is only the donut centre probed by the x-rays or also the surrounding ice-layer ?
- Page 4, line 87 - 89: there are some spelling typos (to the use), (impactable)
- Page 5, line 115: volumetric heating is difficult to understand at this point. The donut shaped coupler is explained after. What is meant by volumetric? What is the probed volume ?
- Page 12, line 317: what is meant by "reactive, low-Z materials". The paper only deals with water, which other low-Z materials do the authors refer to ?
- Melting: The authors describe the absence of Bragg peaks, however, could they identify a smooth halo in the integrated data, which should be visible for liquid water ?

Response to referees

Husband *et al.*, Phase transition kinetics of superionic H₂O ice phases revealed by Megahertz X-ray free-electron laser-heating experiments NCOMMS-24-34499)

We thank both reviewers for taking the time to read and comment on our manuscript. We have included a description of the limitations in temperature determination in this work, as highlighted by reviewer #1, and a detailed reply to the points raised by reviewer #2. The response from reviewer #2 is shown in italics.

Response to reviewer #1

We agree with reviewer 1 that, due to limitations associated with the temperature determination in this work, it is not possible for us to explain discrepancies in the phase lines reported in previous laser-heated DAC experiments. In particular, the fact that the transition to SI ice was reported to occur at higher temperatures in the work by Prakapenka *et al.* (Prakapenka *et al. Nat. Phys.* **17**, 1233–1238 (2021)) compared to the work by Weck *et al.* (Weck *et al. Phys. Rev. Lett.* **128**, 165701 (2022)). We do not claim to address these discrepancies in the manuscript. This is explicitly stated in the manuscript when comparing density discrepancies between our work and previous studies: ‘The possibility that SI-fcc was not observed in our XFEL experiments until ~500 K above the transition temperature is intriguing considering the difficulty in forming SI-fcc, and is consistent with a large energy barrier associated its formation. However, due to the lack of accurate temperature measurements in our XFEL experiments, it is not possible for us to resolve discrepancies between previous work.’ Rather, we comment on the absence of SI-fcc at low pressures, in contrast to Prakapenka *et al.* but in agreement with Weck *et al.*

The results in the manuscript are based on XRD measurements. Although the temperature measurements are consistent with the results, they are not required to support our conclusions. We believe that the limitations in the temperature measurements are already addressed in the manuscript, as outlined below.

-Streaked optical pyrometry (SOP) was used to evaluate the temperature during the X-ray heating experiment. The main limitation of SOP for indirect heating experiments is that we do not directly measure the H₂O temperature, but the temperature of the coupler which indirectly heats the sample via thermal conduction. This is described in the methods section: ‘Due to the low emissivity of H₂O in the temperature region investigated in this work, SOP provided a measurement of the coupler surface temperature at the hot spot, rather than a direct measurement of the sample temperature, which is expected to be lower than that of the coupler.’ In addition, we have been careful to always refer to the coupler temperature when quoting SOP temperatures. In particular, in Fig. 7 (previously Fig. 5) the data points on the phase diagram are identified as corresponding to the coupler temperature in the key, rather than the H₂O temperature. The title of this figure also refers to SOP temperature, rather than H₂O temperature, and the limitation is stated again in the caption: ‘Due to the low emissivity of H₂O, SOP measures thermal emission from the coupler surface, which does not necessarily correspond to the temperature in the H₂O sample.’

-The second limitation of the SOP measurements is related to the temperature oscillations experienced by the coupler (and region of the sample close to the coupler) due to the pump-probe nature of the X-ray heating technique, which cannot be resolved by SOP (shown in Fig. 2, previously Fig. 1). This is stated in the caption of fig. 1: ‘The temporal resolution of the SOP is not sufficient to resolve temperature oscillations during the heating/cooling process; instead, it is sensitive to the hottest part of the cycle where thermal emittance is the brightest.’

-The limitation relating to temporal resolution of the SOP is further discussed in the main text, when we address the fact that SOP temperatures can be high (several thousand K) while the XRD suggests the sample is at room temperature (i.e. the XRD measurements probe the instantaneous temperature of the sample/coupler after cooling, which is lower than the temperature measured by SOP). Specifically, the text reads: ‘Although the diffracted signal from hot Ag (Supplementary Fig. 8) suggests that a portion of the coupler remained close to its melting point after 443 ns, the coupler reflections in the Cu and Au runs remain essentially unchanged from the cold pattern (Supplementary Fig. 9) - despite SOP determining an average coupler temperature (Supplementary Table 3) which is well above the previously-reported lower temperature limit of SI-fcc – suggesting that the coupler cooled significantly between subsequent XFEL pulses. These observations are supported by FEA simulations (Fig. 2(a)) which suggest that the Cu becomes colder than the sample during the cooling process.’

Response to reviewer #2

The paper by R. Husband et al. describes a high-pressure study on superionic ice formed by indirect heating through consecutive X-ray free electron laser pulses. Superionic ice was shown in literature to form at pressures above 20 GPa through different pathways, using e.g. shock compression or optical laser-heating. Two crystal structures of the oxygen sublattice have been found, namely bcc and fcc. The location of the boundary between the two states remained unclear due to different findings. In contrast to previous studies, the current paper is using an interesting new method, where a donut shaped metal coupler is heated through a fs X-ray pulse, simultaneously the ice inside the donut is probed. 300 consecutive pulses continuously increase the temperature of the metal, while the water undergoes a heating cooling oscillation, with overall temperature increase.

The paper is well written and clearly shows the appearance of the bcc structure at pressures below 50 GPa, while fcc is observed only above that threshold. The results are carefully analyzed and critically discussed. The presence of fcc exclusively at higher pressures, supports earlier findings, and resembles an important result, helping to better understand water's phase diagram and in particular the region of superionic ice above ice VII.

I recommend publication of the paper.

Few minor comments:

- In the results and discussion section (page 6) and/or Figure 1 the X-ray beamsize should be clearly provided. Is only the donut centre probed by the x-rays or also the surrounding ice-layer?

We agree with the referee that both the beam size and its position on the coupler is important for these experiments. As mentioned at the beginning of the results and discussion section: 'Data from doughnut-type crucibles were collected with the XFEL beam aligned to the center of the coupler hole so that heating of the coupler was primarily performed by the tails of the focused beam, whereas multiple spots were targeted on nanopowder couplers.' In addition, the methods section specifies that '... the XFEL beam was positioned at the center of the tails hole to maximize the diffracted signal from the H₂O sample. However, the positional jitter in the XFEL beam position meant that the beam position varied from run to run, which could potentially result in different degrees of X-ray heating in subsequent runs due to the large difference in absorption lengths of H₂O and the mid/high-Z couplers.' Three different beam sizes were used for data collection, as noted in the methods section: 'X-rays were focused using a series of CRLs using three different configurations to achieve focal spot sizes of <8 μm, 13 μm, and 26 μm (FWHM), where the beam size was estimated from edge scans using a polished W rod'. The beam size used for individual runs is specified in Table S1.

The volume of ice probed by the XFEL beam depends on the size of the X-ray beam used for that particular run, and the position of the X-ray beam on the coupler due to the jitter of the XFEL beam. Ideally, when the XFEL beam is positioned at the center of the coupler, the focused X-rays will probe the H₂O ice contained within the doughnut, and the surrounding ice/coupler is probed by the tails of the beam. It is not entirely clear what the referee meant by 'the surrounding ice layer'. We have therefore tried to address their question by adding more information on the beam size to the figure captions and text, as suggested. Specifically, we have added an additional sentence to the beginning of the results and discussion section: 'Data were collected using three different X-ray focal spot sizes of <8 μm, 13 μm, and 26 μm (FWHM), which are specified in Table S1.' The beam size used for data collection has also been added to the figure captions of all figures in the main text. We have also specified that the positional jitter is: 'approximately equal to the X-ray focal spot size'.

- Page 4, line 87 – 89: there are some spelling typos (to the use), (impactable)

We thank the referee for pointing out the typos in section of the manuscript, which have now been corrected. The text now reads: 'Although CO₂ laser heating provides more spatially uniform heating as the laser is directly absorbed by the sample, CO₂ lasers typically exhibit large power fluctuations at microsecond timescales due to the use of pulse width modulation to control the average laser power. In addition, direct CO₂ laser heating of H₂O is not possible above ~60 GPa due to the reduced absorption of $\lambda \sim 10 \mu\text{m}$ radiation above this pressure.'

- Page 5, line 115: volumetric heating is difficult to understand at this point. The donut shaped coupler is explained after. What is meant by volumetric? What is the probed volume?

The word volumetric was initially included in the sentence: ‘the near-instantaneous energy deposition in the sample during single pulse irradiation produces ultrafast (sub-ns) volumetric heating followed by rapid cooling and relaxation due to heat dissipation’. Here, the word ‘volumetric’ was intended to highlight one of the main advantages of direct X-ray heating, which is independent of the coupler concept. Specifically, the fact that typical X-ray absorption lengths at 18 keV (e.g. $\sim 4000 \mu\text{m}$ for H_2O at 40 GPa, $\sim 5 \mu\text{m}$ for Au, and $\sim 25 \mu\text{m}$ for Cu) are comparable with (or significantly longer than) the sample thickness within the DAC, so that the absorbed energy is deposited deep within the sample (i.e. approximately the entire volume of the irradiated region of the sample is heated, depending on the specific material and thickness). This contrasts with IR laser heating (typically used to generate high temperatures in DACs), in which only the surface of the sample is heated due to the short penetration depth of IR radiation (10s of nm in metals), which can result in significant temperature gradients within the sample. This has previously been listed as one of the main advantages of direct X-ray heating in comparison to traditional heating methods [e.g. in Liermann *et al.* *J. Synchrotron Rad.* **28**, 688–706 (2021)].

However, we agree with the referee that the use of the word ‘volumetric’ in this context may be confusing – in particular, as it may be assumed to refer to the indirect X-ray heating approach, in which the H_2O sample is primarily heated via thermal conduction from the hot coupler. We have therefore removed the word ‘volumetric’ from this sentence to avoid confusion.

- Page 12, line 317: what is meant by “reactive, low-Z materials”. The paper only deals with water, which other low-Z materials do the authors refer to ?

The term ‘reactive, low-Z materials’ was initially included in the sentence: ‘In conclusion, an indirect XFEL heating technique was used to investigate the structural properties of SI ice, demonstrating that stepwise XFEL heating is a viable method to study reactive, low Z materials at high pressures and temperatures.’ We initially intended this sentence to highlight the potential of indirect XFEL heating to study a range of reactive, low Z materials in future studies, but realise that the wording may be confusing. We have therefore modified this sentence to read: ‘In conclusion, an indirect XFEL heating technique was used to investigate the structural properties of SI ice, demonstrating that stepwise XFEL heating is a viable method to study low Z materials such as H_2O at high pressures and temperatures.’

- Melting: The authors describe the absence of Bragg peaks, however, could they identify a smooth halo in the integrated data, which should be visible for liquid water ?

We agree with the referee that we ideally would have observed a diffuse scattering signal (i.e. smooth halo) from liquid H_2O after it melted. However, diffuse scattering was not clearly observed in these runs. We attribute this to the weak scattering signal of H_2O combined with the limited azimuthal coverage of the AGIPD detector. We note that the identification of diffuse scattering from liquid H_2O in high temperature DAC experiments relies on the observation of a weak H_2O signal which lies on top of the Compton scattering background signal from the diamond anvils. Previous laser-heating synchrotron XRD experiments on H_2O used a multichannel collimator (MCC) system to reduce the diamond Compton signal and enhance the molten signal (e.g. Queyroux *et al.* *Phys. Rev. Lett.* **125**, 195501 (2020). and Weck *et al.* *Phys. Rev. Lett.* **128**, 165701 (2022)). The H_2O molten signal was very weak in studies which did not employ a MCC (e.g. Fig. 2b in Prakapenka *et al.* *Nat. Phys.* **17**, 1233–1238 (2021).) An MCC was not used in our work, as it is not compatible with single pulse XFEL experiments. The difficulty in identifying the weak liquid signal above the Compton scattering signal is further complicated in XFEL experiments by the fluctuating X-ray intensity during the X-ray pulse train, which means that the Compton scattering background signal fluctuates from frame-to-frame. In general, diffuse scattering signals in XFEL heating experiments performed using the DAC platform has found to be very weak (at best) even for high Z samples, as confirmed by the beamline staff at HED. Finally, we note that, even if diffuse scattering was observed, it would be difficult to distinguish diffuse scattering of H_2O from that of the high Z coupler, which is likely to be partially molten in these runs.

The absence of diffuse scattering in these runs is explicitly stated in the manuscript: ‘Although no evidence of diffuse scattering from the fluid was observed due to the weak scattering power of H_2O and the limited azimuthal coverage of the detector, the possibility that the disappearance of the H_2O solid signal originated from chemical reactivity or sample loss was ruled out by the strong ice VII signal and absence of chemical reaction products in the first pulse pattern of the subsequent run (Supplementary Fig. 12).’